# Heterogeneous Multitype Fleet Green Vehicle Path Planning of Automated Guided Vehicle with Time Windows in Flexible Manufacturing System

**Jia Gao [1], Xiaojun Zheng [2,*], Feng Gao [2], Xiaoying Tong [2] and Qiaomei Han [3]**

[1] School of Locomotive and Rolling Stock Engineering, Dalian Jiaotong University, Dalian 116021, China; gaojia_acade@163.com

[2] School of Mechanical Engineering, Dalian Jiaotong University, Dalian 116021, China; m13179292992@163.com (F.G.); txy@djtu.edu.cn (X.T.)

[3] Department of Electrical and Computer Engineering, Western University, London, ON N6A5B9, Canada; qhan42@uwo.ca

\* Correspondence: zhengxj@djtu.edu.cn

**Abstract:** In this study, we present and discuss a variant of the classical vehicle routing problem (VRP), namely the heterogeneous multitype fleet green automated guided vehicle (AGV) routing problem with time windows (HFGVRPTW) applied in the workshops of flexible manufacturing systems (FMS). Specifically, based on the analysis of AGV body structure and motion state, transport distance and energy consumption are selected as two optimization objectives. According to the characteristics and application context of the problem, this paper designs a hybrid genetic algorithm with large neighborhood search (GA-LNS) considering the farthest insertion heuristic. GA-LNS is improved by increasing the local search ability of genetic algorithm to enhance the solution optimal quality. Extensive computational experiments which are generated from Solomon's benchmark instances and a real case of FMS are designed to evaluate and demonstrate the efficiency and effectiveness of the proposed model and algorithm. The experimental results reveal that compared with using the traditional homogeneous fleet, the heterogeneous multitype AGV fleet transportation mode has a huge energy-saving potential in workshop intralogistics.

**Keywords:** vehicle routing problem; heterogeneous multitype fleet; AGV; time windows; energy consumption; hybrid genetic algorithm

## 1. Introduction

Industry 4.0, as a revolutionary industrial paradigm, has given today's manufacturing industry a new technical environment and competition context. Nowadays, global manufacturing industries are transforming from mass production to multi-variety, small-batch and customized production modes [1,2]. FMS is beneficial to improving production system efficiency and increasing production process flexibility. Thus, FMS has been rapidly constructed in recent decades. As a flexible material handling device inside FMS, AGV can improve the production flexibility, it has gradually replaced manual handling with an uncompromised performance in recent years [3,4].

Nowadays, energy conservation and green manufacturing have already become a trend of manufacturing industry [5]. Nevertheless, the current energy-saving research is still focused on automated workstations or production equipment, and the optimization of the material handling system in FMS is insufficient. When the AGV performs handling tasks, it is usually powered by loaded batteries which have to be charged every 8–10 h [6]. Therefore, the task execution will be interrupted by charging, which will cause delays of the production plan. Thus, how to effectively schedule and plan energy-saving paths is paramount to AGV, which can not only save energy-consumption of the entire FMS, but also improved the stability of the material handling system.

In the workshop of modern FMS with many customized products and large volume of materials it is hardly to achieve better green economic benefits by using homogeneous AGV fleet transportation. In contrast, the use of heterogeneous multitype AGVs fleet, which differ in capacities and energy consumption, has multiple advantages such as improving vehicle utilization, relieving the pressure of traffic flow in workshop, and reducing workshop intralogistics cost to a certain extent [7].

HFGVRPTW as a variant of the classical vehicle routing problem with time windows (VRPTW) was proposed. Furthermore, this research is considering the increasing attention to the sustainable of manufacturing process and the application context of energy-saving workshops, the energy-saving path planning of the heterogeneous multitype fleet AGVs was studied. The difficulty of this research is to coordinate and optimize economic and environmental benefits through scientific planning of the vehicle fleet size, vehicle types and traverse path under the time windows and capacity constraints.

Among the numerous previous studies on green vehicle routing problem (GVRP), the energy consumption or fuel consumption is closely relevant to the real-time payload weight of vehicles body [8–10]. Nevertheless, as shown as Figure 1, reference [11] through implemented simulation model and statistical analysis of experimental data, it is confirmed that the servo motor output power of the AGV body rises with increasing payload weight, but this power fluctuation phenomenon shows only minor deviations. Considering this energy consumption characteristic of AGVs, based on reasonable assumptions and quantitative analysis of the motion state of AGVs transportation process, an AGV path planning model was established with two performance indicators of transportation distance and energy consumption. Then, by solving this model and analyzing the experimental results, we hope to reveal superiority of the proposed model and algorithm over the homogeneous fleet transportation.

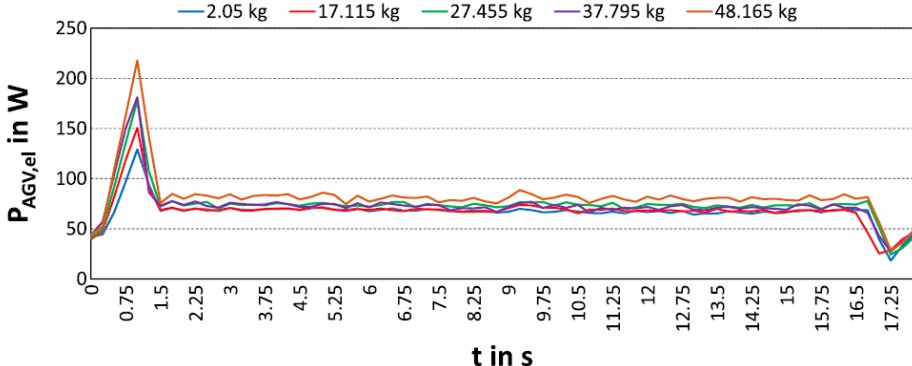

**Figure 1.** Influence of the payload weight on the electrical power [11].

The purpose of this paper is to minimize transportation distance and energy-consumption for the heterogeneous multitype AGVs fleet to complete the loading task. Our main contributions in this research can be summarized as follows:

- We analyze the motion state and vehicle body structure of AGVs, so that the established mathematical model is more suitable for the actual application.
- We propose a hybrid genetic algorithm with large neighborhood search. Furthermore, by enhancing the local search ability of genetic algorithm and optimization solution can be ameliorated.
- We empirically demonstrate that the application of heterogeneous multitype AGVs fleet in FMS can improve the performance of the average utilization resources and energy efficiency.

The rest of the paper is organized as follows. Section 2 reviews the related literature on the VRP, AGV path planning and outlines the research gaps. Section 3 established mathematical programming models and analyzes the motion and energy consumption of AGV transport. Furthermore, we proposed a hybrid improved algorithm GA-LNS for

energy-saving HFGVRPTW in Section 4. To verify the model effectiveness and the robustness of GA-LNS, we tested it on a set of benchmark instances and an actual environment example. In Section 5, we present the results of those computational tests and investigation the factors affecting the optimization objectives of model. Finally, conclusions and further research directions are stated in Section 6.

## 2. Literature Review

In recent years, the green vehicle routing problem (GVRP) in workshop intralogistics has been an important issue [12,13]. Relevant studies are concentrated on the following two aspects: GVRP and heterogeneous multitype fleet vehicle routing problems with time windows (HFVRPTW). Hence, there is need an organized literature review that seeks the advancement in GVRP application in FMS. To make the literature overview clearer, Table 1 summarizes the above works. To conclude, although there are many GVRP variants, there is still much room for improvement.

### 2.1. Green Vehicle Route Problem

In 2016, the International Energy Agency pointed out that transport activities were the second largest contributor to $CO_2$, accounting for 24% of global $CO_2$ emissions [14]. The concept of GVRP is proposed by considering the eagerness to reduce carbon emissions and realize green logistics [15]. GVRP is a variation of the classical VRP, belongs NP-hard problem. The basic GVRP can be defined thus: under the premise that the location of the loading point and the vehicle loading capacity are known, the number of departures as well as the transport path of the vehicle are scientifically planned to meet the loading requirements under various constraints while minimizing energy consumption. Moreover, the established hypothesis is using a homogeneous vehicle fleet to perform loading tasks and all loading points can only be served once [16].

It seems that GVRP only adds energy consumption indicator to the optimization goal, which is not essentially different from the classical VRP. In fact, the mathematical model and optimization solution of GVRP is much more complicated than classical VRP for the energy consumption of the vehicle is affected by motion state, characteristics parameters and real-time speed. Besides, most VRP research has commonly considered that transport cost was mainly determined by the length of travel distance. In fact, shortening travel distance cannot guarantee cost reduction because of different vehicle speed, motion state and penalty cost caused by the time window constraint.

For AGV fleets in FMS, electrical energy consumption generates carbon emissions and pollution as power supply [17]. Moreover, electricity cost is a significant part of the transport cost which is the basis of GVRP study. The existing literatures have conducted in-depth research on the impact of vehicle time-varying speed and average speed on vehicle energy consumption. Furthermore, the comprehensive modal emissions model (CMEM) is a parameterized approach based on energy consumption and carbon emission production. CMEM is suitable for the energy consumption and emission prediction of individual vehicles or an entire fleet of multispeed types [18]. Yang Y et al. [7] constructed a heterogeneous fleet GVRP model with vehicle number constraints. This model is taking the minimization of carbon emissions as the optimization goal, using CEME as a fuel consumption measurement, considering the average vehicle speed during haul. However, in the actual transport, the vehicle speed is fluctuant. In a similar study, Fuskasawa et al. [19] use CMEM as a fuel consumption measurement model. In this scheme, speed is used as a continuous decision variable to make the vehicle drive at the optimal speed on each road section. A mathematical model with vehicle numbers and time windows is constructed.

In line with sustainable development, electrical vehicles are gradually replacing fuel vehicles as the focus of GVRP studies. The electric VRP (EVRP) has become the main trend in VRP-related issues. Compared with conventional fuel-powered vehicles, battery-powered electric vehicles take a long time to charge. For this trait, the energy-consumption model and the battery management technology of electric vehicles can adapt to the trans-

portation requirements. Typically, Yiyong Xiao et al. [20] investigated the EVRP with time windows (EVRPTW) by considering the speed-varying travel range and electricity consumption throughout the vehicle tours. Seyed F. et al. [21] presents a new measure model and solution for the scheduling problem of HFVRP which considers the fuel consumption rate and customers' satisfaction level.

On the other hand, Ugur B. et al. [22] integrated the electric vehicle idle time and recharging time into the mathematical model. Qazi S. et al. [23] explored how different routing techniques for the battery management of AGVs effect the efficiency performance of a manufacturing facility. Mengting Zhao and Yuwei Lu [24] investigated a real-world EVRP combining multiple features of VRP which is devised to decrease the total operational cost. However, they ignored the change of the motion state of vehicle haul, which is contrary to the actual situation. Yubang Liu et al. [25] investigated the task scheduling of AGVs by taking the charging task and the time-varying speed. Considering the minimize number of AGVs used and the amount of electricity consumption as optimization goal. Macrina et al. [26] proposed a mixed vehicle fleet composed of electric and conventional VRP with partial battery recharging and time windows.

From aforementioned studies, the appropriateness of metrics for vehicles energy consumption is electric consumption and fuel consumption. They are closely related to such factors as speed, vehicle characteristic parameters and travel distance. Besides, current GVRP studies are mainly focused on the multiple vehicles route planning, scheduling for outdoor logistics. The GVRP within FMS, which is different from the outdoor environment, has not been explored in-depth. Although some GVRP studies were mentioned, the energy consumption characteristics of AGV have not been fully considered, and the presented models were either based on steady speed during the whole transport process or based on battery management. Based on current research status, we proposed a model which can directly calculate the AGV electric consumption in the entire transport haul with various vehicle motion state.

### 2.2. Heterogeneous Multitype Fleet Vehicle Routing Problems with Time Windows

The use of heterogeneous multitype fleet transportation can reduce costs while ensuring transportation efficiency. Reference [27] demonstrated that for urban logistics, a heterogeneous fleet is superior to a homogeneous fleet. In FMS, the production scheduling system often restricts the loading time of each loading point with a specific time interval [22]. Because of HFVRPTW practical relevance and intrinsic complexity, it has been a research topic for algorithm innovation and solving practical routing related problems.

Based on mathematic models and environment representation, numerous paths search algorithms have been proposed in existing studies. They can be categorized into two types: conventional algorithms and meta-heuristic algorithms. The former consists of branch-and-price (BAP) algorithm and mixed integer linear programming algorithm. Ece Y. et al. [28] considered the limited number of vehicles during the distribution process, orders need to decrease the total travelling time. In this situation, some of customer orders tardy. To minimize the total delay time, a mixed integer programming formulation to model this problem is proposed. Wei Q. et al. [29] established a mixed-integer linear programming model to deal a practical HFVRP to minimize the maximum routing time of vehicles. Yang Yu et al. [7] proposed a dynamic BAP algorithm to precisely solve HFGVRPTW. Since the exponential explosion problem in the process of solving the optimal path is unavoidable for the conventional algorithm, caused its real-time performance is worse, so it is only suitable for solving small-scale VRP problems.

In contrast, meta-heuristic algorithms are widely used to solve variant VRP problems due to their extremely global search capabilities. Ugur B. et al. [22] proposed an EVRPTW framework which considered multiple depots and a heterogeneous vehicle fleet. To deal with the proposed problem, a series of neighborhood search operators were developed. Nasser R.S. et al. [30] proposed an adaptive multi-local search algorithm to solve HFVRPTW with two-dimensional loading constraints. Considering the HFVRP with simultaneous

pickup and delivery, Napoleão N. et al. [31] proposed a randomized algorithm to tackle it. However, the search efficiency of traditional heuristic algorithm is unsatisfactory, because it is easy to fall into local optimal and lead to hardly achieve global optimization. Thus, most existing research uses multipoint metaheuristic algorithms to solve VRP and its variant problem. Vahid B. et al. [32] addressed HFVRPTW with multiple hard prioritized. Due to the NP-hard problem, a bionic algorithms called binary artificial bee colony algorithm is developed to solve the problem. Zhong et al. [33] devised a particle swarm optimization algorithm to solve a mixed based on path optimization and integrated scheduling. Zeng H et al. [34] investigated an improved genetic algorithm on AGVs path planning which apply three-exchange crossover heuristic operators to produce more optimal offspring. The improvement and combination of two algorithms are more popular nowadays. Hou D. et al. [35] was taking the minimizing distribution cost as the optimization solution target, solve it by using a hybrid genetic algorithm. Puca et al. [36] address HFVRP, in their approach, a hybrid meta-heuristic which combines an iterated local search with variable neighborhood descend is proposed. The efficiency of the algorithm is tested by large-scale well-known benchmark instances.

The transportation network in the FMS is usually abstracted as a topological map and the optimal paths are planned and searched based on it. Besides that, AGV is usually treated as a particle. This simplification applies for most path planning, but this method may affect the evaluation accuracy of transportation distance and energy consumption [36]. For example, when the AGV is turning through a path intersection of transport, the steering energy consumption should be combined with the force of its steering mechanism and speed change. Meanwhile, if the AGV is still abstract as a particle, it is not accurate to analyze the energy consumption of the AGV movement. Furthermore, with the increase of the path planning problem's scale and corresponding constraints, it is a tough job to obtain the optimal solution within a suitable time. For this reason, multipoint metaheuristic algorithms have a strong search efficiency to deal with uncertainties of the environment such as FMS, therefore in this study we proposed GA-LNS.

**Table 1.** Tabular form of the related reference.

| Reference | Algorithm | Time Windows | Objective | | | Heterogeneous Vehicle Fleet |
|---|---|---|---|---|---|---|
| | | | Energy Consumption | Cost | Other | |
| Fuskasawa et al. [19] (2016) | Exact | √ | | | √ | |
| Marcel Turkensteen [18] (2017) | - | | √ | | | |
| Zeng et al. [34] (2017) | Heuristic | | | | √ | |
| Zhang et al. [17] (2018) | Heuristic | | √ | | | |
| Seyed F. et al. [21] (2018) | Heuristic | √ | √ | | | √ |
| Macrina et al. [26] (2018) | Heuristic | | | | | |
| Yang et al. [7] (2019) | Exact | √ | | √ | | |
| Xiao et al. [20] (2019) | Exact | √ | √ | | | |
| Qazi S. et al. [23] (2019) | Heuristic | | | | √ | |
| Meng and Lu [24] (2019) | Heuristic | √ | √ | | | √ |
| Liu et al. [25] (2019) | Hybrid | | | | √ | |
| Napoleão N. et al. [31] (2019) | Heuristic | | | √ | | √ |
| Vahid B. et al. [32] (2019) | Heuristic | √ | | √ | | √ |
| Puca et al. [36] (2019) | Heuristic | √ | | √ | | √ |
| Nasser R.S. et al. [30] (2020) | Heuristic | | | √ | | √ |
| Zhong et al. [33] (2020) | Hybrid | √ | | | √ | |
| Ece Y. et al. [28] (2021) | Exact | | | | √ | √ |
| Wei Q, et al. [29] (2021) | Exact | | | √ | | √ |
| Ugur Bac and Mehmet Erdem [22] (2021) | Heuristic | √ | | | √ | √ |
| Hou D. et al. [35] (2021) | Heuristic | √ | | | | √ |
| Our study | Hybrid | √ | √ | | | √ |

## 3. Mathematical Model

### 3.1. Problem Description

The mathematical notations for the presented HFGVRPTW are given in Table 2.

**Table 2.** Notations.

| Notation | Description |
|---|---|
| **Sets** | |
| A | Set of graph edges |
| G | Undirected graph representing a flexible manufacturing workshop |
| M | Set of AGV types |
| N | Set of loading points |
| O | Set of all AGVs |
| S | Ordered node set representing a path |
| V | Node set of a graph |
| **Indices** | |
| $i, j$ | Node, $i, j \in V$ |
| $k$ | Acceleration stage index, $k \in \{1, 2, \ldots, p\}$ |
| $l$ | Deceleration stage index, $l \in \{1, 2, \ldots, q\}$ |
| $m$ | AGV type, $m \in \{1, 2, \ldots, |M|\}$ |
| $n$ | Number of nodes in the set of $N$, $n \in \{1, 2, \ldots, n\}$ |
| $o$ | Any AGV in the AGV set $O$, $o \in \{1, 2, \ldots, O\}$ |
| **Parameters** | |
| $a_{acc}$ | Acceleration of the accelerated motion [m/s$^2$] |
| $a_{dec}$ | Acceleration of the decelerated motion [m/s$^2$] |
| $d_{ij}$ | Distance between node $i$ and , also, $i \neq j$ [m] |
| $D$ | Total transport distance between nodes [m] |
| $D_{acc}$ | Total acceleration displacement [m] |
| $D_{dec}$ | Total deceleration displacement [m] |
| $D_{usm}$ | Uniform straight motion distance [m] |
| $D_{urm}$ | Uniform turning motion distance [m] |
| $E_{acc}^{k}$ | Energy consumed to support the accelerated motion in the $kth$ acceleration stage [J] |
| $E_{alm}$ | Energy consumption caused by all accelerated motion [J] |
| $E_{sm}$ | AGV energy consumption caused by standby motion [J] |
| $E_{slm}$ | Total AGV energy consumption caused by standby motion [J] |
| $E_{total}$ | Total transport energy consumption of one AGV [J] |
| $E_{ulm}$ | Energy consumed by uniform motion in all uniform motion [J] |
| $E_{urm}$ | Total AGV energy consumption caused by uniform turn motion [J] |
| $E_{usm}$ | Total AGV energy consumption caused by uniform straight motion [J] |
| $F_m$ | Motor output driving force of vehicle type $m$ [N] |
| $l_n$ | Loading time at node $n$, and the loading time at node 0 is set to $(0, +\infty)$ [min] |
| $L_m$ | AGV length of vehicle type $m$ [m] |
| $[LT_n, RT_n]$ | Time window of the node $n$ |
| $Q_m$ | Time window of the node $n$ [kg] |
| $q_n$ | The material weight of the point $n$ [kg] |
| $n_T$ | Number of nodes that an AGV turns through |
| $o_m$ | Any AGV of type $m$ |
| $R_m$ | AGV turning radius of vehicle type $m$ [m] |
| $R_1, R_2, R_3, R_4$ | Turning radius of AGV wheel [m] |
| $t_{acc}^{k}$ | Time of the $kth$ acceleration stage [s] |
| $t_{dec}^{l}$ | Time of the $lth$ deceleration stage [s] |
| $t_{ij}$ | Total execution time of the one AGV from node $i$ [s] |
| $T_n$ | The start time of the loading task at load point $n$ |
| $T_{total}$ | Total execution time of one AGV [min] |
| $v_{0k}$ | AGV initial velocity in the $kth$ acceleration stage [m/s] |
| $v_{0l}$ | AGV initial velocity in the $lth$ deceleration stage [m/s] |

**Table 2.** *Cont.*

| Notation | Description |
|---|---|
| $v_r$ | Uniform turning motion speed [m/s] |
| $v_s$ | Uniform straight motion speed [m/s] |
| $v_{tk}$ | AGV terminal velocity in the *kth* acceleration stage [m/s] |
| $v_{tl}$ | AGV terminal velocity in the *lth* deceleration stage [m/s] |
| $W_m$ | AGV width of vehicle type *m* [m] |
| $\eta_m$ | Overall power factor of driving motors of vehicle type *m* |
| **Decision Variable** | |
| $x_{ijo_m}$ | $x_{ijo_m} = \begin{cases} 1, & \text{if AGV } o_m \text{ travels from node } i \text{ to node } j \\ 0, & \text{otherwise} \end{cases}$ |

HFGVRPTW can be defined on a complete directed graph $G = \{V, A\}$ with a set of nodes $V = \{0\} \cup N$ and a set of edges $A = \{(i, j)|i, j \in V, i \neq j\}$. Node 0 is the distribution center of the heterogeneous multitype fleet AGVs, $N = \{1, 2, \ldots, n\}$ is the set of loading point. An unlimited multitype fleet of heterogeneous AGVs is available at the workshop, perform material loading and handling tasks for the entire workshop. The transit distance and time on edge $(i, j)$ are defined as $d_{ij}$ and $t_{ij}$, respectively. The distribution center is composed of $M$ different types of AGVs. Each AGV type $m = \{1, 2, \ldots, |M|\}$ has a maximal payload $Q_m$, a motor output driving force $F_m$. $O_m$ is a collection of type $m$ AGVs, $o_m$ is any AGV of type $m$. Each loading point $n \in N$ has the quality of material $q_n$, a loading time $l_n$, and a time window $[LT_n, RT_n]$, where $LT_n$ and $RT_n$ are the earliest and latest allowable start loading times, respectively. Each AGV should start the perform loading tasks of a loading point within the given time windows. If the AGV arrives earlier than $LT_n$, a waiting energy consumption of the waiting time is incurred. If it is later than $RT_n$, the loading task execution fails. In Figure 2, two types of AGVs, i.e., Vehicle 1 and Vehicle 2, are used fulfill loading tasks of the workshop. They need to start from the distribution center and return it after serving a certain number of loading points. Therefore, with a given heterogeneous multitype AGVs fleet and a given set of transport tasks, the overall problem is to determine the optimal vehicle scheduling as well as AGVs path planning. The optimization objective is to minimize the sum of transport distance and energy consumption.

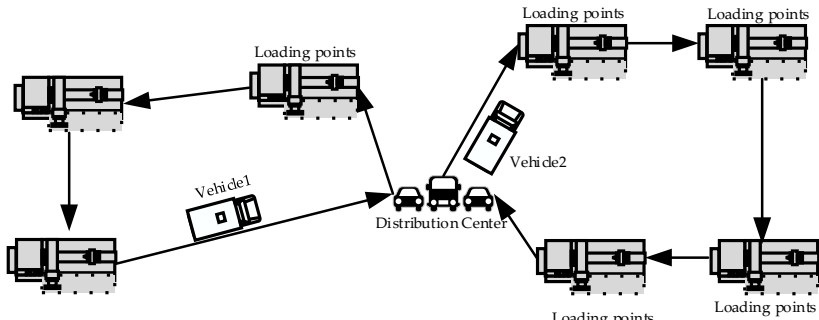

**Figure 2.** Illustration for the proposed problem.

Additionally, the following assumptions are made for the model formulation. (1) All loading points must be loaded. (2) The wheels do not slip when the AGV is in motion. (3) The locations of the workstations in the workshop are fixed and known. (4) All AGVs are parked in the starting point until scheduling commands are assigned. Once the material handling process assigned, it may leaded some traffic problems. In this study, AGVs prevent traffic problems by relying on the infrared obstacle avoidance module on their vehicle body [5].

*3.2. Energy Consumption Analysis*

　　When the AGV performs the handling work in the flexible manufacturing workshop, it is usually powered by its own load battery. During the driving process of AGV, there are mainly four kinds of motion states, i.e., standby motion, uniform motion, decelerated motion, and accelerate motion. In addition, when the AGV is decelerating, the output power of the driving motor generally drops sharply, even to zero. As such, the energy consumption to maintain the deceleration movement is ignored in the energy consumption analysis, but the deceleration movement time is one of the components of $T_{total}$, and still needs to be considered in the time analysis. Correspondingly, the total AGV transport energy consumption $E_{total}$ is mainly composed of three parts: $E_{slm}$, $E_{ulm}$, and $E_{alm}$.

3.2.1. Standby Energy Consumption Analysis

　　During the production process of FMS, each loading point has its corresponding time window. For this reason, time analysis of the entire driving process of the AGV is necessary.

　　Based on previous assumptions, $t_{ij}$ is the sum of the acceleration time, deceleration time and uniform motion time of the AGV driving time from loading point $i$ to $j$. Considering that the acceleration and deceleration motion only occurs on the straight path that it is about to turn. Furthermore, the AGV turns at uniform speed. Therefore, the $t_{ij}$ for a single AGV to perform its corresponding loading task can be expressed as:

$$t_{ij} = \sum_{k=1}^{p} t_{acc}^{k} + \sum_{l=1}^{q} t_{dec}^{l} + \frac{D_{urm}}{v_r} + \frac{D_{usm}}{v_s} \tag{1}$$

　　For given any transport path S, the number of acceleration phase, deceleration phase and the distance of uniform straight motion $D_{usm}$ can all be determined. Now suppose that for path graph G contains total of $p$ acceleration phases, $q$ deceleration phases, the time used in the *kth* acceleration phase and the deceleration time of the *lth* can be expressed as below:

$$t_{acc}^{k} = \frac{v_{tk} - v_{0k}}{a_{acc}}, k = 1, 2, \ldots, p \tag{2}$$

$$t_{dec}^{l} = \frac{v_{tl} - v_{0l}}{a_{dec}}, l = 1, 2, \ldots, q \tag{3}$$

Traverse all nodes distance $D$ can be expressed as that:

$$D = \sum_{i=1}^{n} \sum_{j=1}^{n} d_{ij} x_{ij} \tag{4}$$

　　Considering the uniform motion will appear in the straight and turning phases, the uniform straight motion distance $D_{usm}$ can be calculated as:

$$D_{usm} = D - D_{acc} - D_{dec} \tag{5}$$

　　Further, the total acceleration displacement $D_{acc}$ and deceleration displacement $D_{dec}$ can be obtained as:

$$D_{acc} = \sum_{k=1}^{p} [v_{0k} t_{acc}^{k} + \frac{1}{2} a_{acc} \left( t_{acc}^{k} \right)^2] \tag{6}$$

$$D_{dec} = \sum_{l=1}^{q} [v_{0l} t_{dec}^{l} + \frac{1}{2} a_{dec} \left( t_{dec}^{l} \right)^2] \tag{7}$$

　　The uniform turning motion distance $D_{urm}$ which determined by the number of AGV turns can be formulated as follows:

$$D_{urm} = \frac{1}{2} \pi R_m n_T \tag{8}$$

From the above path analysis, the energy consumption of AGV can be calculated by:

$$E_{slm} = E_{sm} \times [LT_n - t_{ij}] \tag{9}$$

3.2.2. Energy Consumption Analysis of Uniform Motion

Regarding uniform motion will occur both in the straight and turning sections of all AGVs, i.e., uniform motion driving energy consumption $E_{ulm}$ consists of two parts: the uniform straight motion consumption $E_{usm}$ and the uniform turn motion consumption $E_{urm}$. Therefore, it is necessary to analyze the AGV's straight and turning transport distance to determine the total path length of AGV in topological map. When a transport path is given, the straight distance and the number of turn points will be determined, denote them as $D_{usm}$ and $n_T$, respectively. Constrained by the car body structure, the turning radius of the inner and outer wheels of the AGV is different during turning process. A schematic diagram of a common four-wheel AGV turning radius is shown as Figure 3, which can be calculated with the uniform turning motion distance $D_{urm}$ [5].

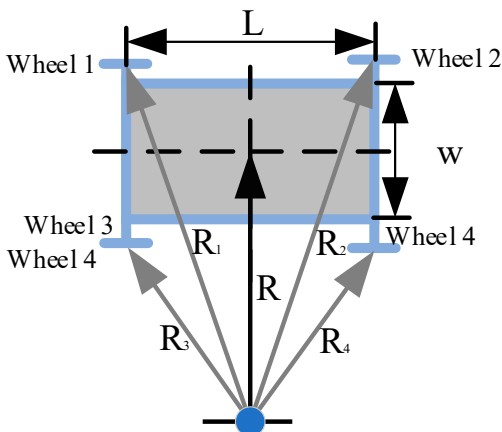

**Figure 3.** Schematic diagram of four-wheel AGV turning radius.

The turning radius of each wheel in the AGV car body can be expressed as [5]:

$$\begin{cases} R_1 = R_2 = \sqrt{\left(R_m + \frac{1}{2}W_m\right)^2 + \left(\frac{L_m}{2}\right)^2} \\ R_3 = R_4 = \sqrt{\left(R_m - \frac{1}{2}W_m\right)^2 + \left(\frac{L_m}{2}\right)^2} \end{cases} \tag{10}$$

In this study, we assume that the AGV drive motors are evenly distributed among four wheels, the total energy consumption during the turning phase $E_{urm}$ can be represented by the follow formula [5]:

$$E_{urm} = \frac{\left[F_m \times \frac{\pi}{2} \times \frac{(R_1 + R_2 + R_3 + R_4)}{4}\right]}{\eta_m} \times n_T \tag{11}$$

Similarly, the sum of energy consumption in all straight phase $E_{usm}$ can be obtained as below:

$$E_{usm} = \frac{\sum\limits_{i,j=1}^{n} F_m \times D_{usm}^{ij}}{\eta_m} \tag{12}$$

Consequently, $E_{ulm}$ can be obtained by:

$$E_{ulm} = E_{urm} + E_{usm} \tag{13}$$

### 3.2.3. Energy Consumption Analysis of Acceleration Motion

Because of the acceleration movement only occur to the straight line about to turn, at this time, the energy consumption in the *kth* acceleration movement is:

$$E_{acc}^k = \frac{F_m^k \times [v_{0k}t_{acc}^k + \frac{1}{2}a_{acc}(t_{acc}^k)^2]}{\eta_m} \tag{14}$$

Thus, $E_{alm}$ can be calculated by:

$$E_{alm} = \sum_{k=1}^{p} E_{acc}^k \tag{15}$$

### 3.3. Optimization Objectives and Constraints

The combined optimization objective function of this study can be formulated as below:

$$\begin{cases} \text{Minimize}[D] \\ \text{Minimize}[E_{total}] \end{cases} \tag{16}$$

The constraints are as follows:

$$\sum_{i \in N} x_{i0o_m} = \sum_{j \in N} x_{0jo_m} \leq 1, \forall m \in M, \forall o_m \in O_m \tag{17}$$

$$\sum_{i \in \{0\} \cup N} \sum_{j \in N} x_{ijo_m} q_n \leq Q_m, \forall m \in M, \forall o_m \in O_m, \forall n \in N \tag{18}$$

$$\sum_{j \in N} \sum_{o_m \in O_m} x_{0jo_m} \leq |O_m|, \forall m \in M \tag{19}$$

$$\begin{cases} \sum_{j \in \{0\} \cup N} \sum_{o \in O} \sum_{o_m \in O_m} x_{ijo_m} = 1, \forall i \in N \\ \sum_{i \in \{0\} \cup N} \sum_{o \in O} \sum_{o_m \in O_m} x_{ijo_m} = 1, \forall j \in N \end{cases} \tag{20}$$

$$T_n + \sum_{i \in \{0\} \cup N} \sum_{j \in \cup N} t_{ij}x_{ijo_m} + \sum_{i \in \{0\} \cup N} \sum_{j \in \cup N} l_n x_{ijo_m} \leq RT_n, \forall m \in M, \forall o_m \in O_m \tag{21}$$

$$\sum_{i \in S} \sum_{j \in S} x_{ijo_m} \leq |S| - 1, S \subseteq N, \forall m \in M, \forall o_m \in O_m \tag{22}$$

In above model, objective (16) is to minimize total traverse distance as well as minimize total AGV drive energy consumption. Constraint (17) represents each AGV has only one distribution path and must start and end at the distribution center. Constraint (18) represents the capacity limit of the AGV. Constraint (19) indicates that the number of AGV leaving the distribution center does not exceed the total number of the AGVs that it can be used to ensure that the planned feasible path meets the actual logistics needs of FMS. Constraint (20) ensures that each loading point is traversed only once by one AGV. Constraint (21) restricts the time when the AGV completes the loading task of one loading point not to exceed the right time windows $RT_n$. Constraint (22) is to eliminate the sub-loop constraint.

## 4. Algorithm Design

Considering the complexity of HFGVRPTW, based on the large-scale neighborhood search (LNS) and genetic algorithm (GA), we devise a metaheuristic algorithm to solve it. The pseudocode of this algorithm is given as Algorithm 1.

The LNS was first proposed by Paul Shaw [25]. Its basic solution process is to start with the initial solution continuously optimize the solution through removal and insertion operators to generate new flexible solutions. GA is a heuristic algorithm which searches

for optimization solutions by simulating the natural evolution process in biology. It has the characteristics of better global search performance and strong robustness. GA is to generate a new generation of population through repeatedly select, crossover, and mutate. Nevertheless, the algorithm also has the disadvantages of slow convergence rate due to insufficient local search ability and easy falling into local optimization. In recent years, there have been many improvements in GA. The most common improvement directions used are [24]:

- To combine with other algorithms to improve the neighbor structures of the feasible solution of GA, and enhance the local search ability of GA while maintaining the diversity of the population;
- To optimize the evolutionary steps in GA in order to reduce damage to good genes.

Combining the advantages of GA and LNS, GA-LNS is specialized to the HFGVRPTW is developed to solve the proposed problem.

---

**Algorithm 1:** Pseudocode of GA-LNS.

---

**Parameters:**
*pop_size* : population size;
*MAX_gen* : maximum number of iterations
$N_k = \{N_1, N_2, \ldots, N_l\}$: neighborhood solution　$N_l$　is the *lth* feasible neighborhood solution;
$F_{N_l}$ : fitness function of the constructed feasible neighborhood solution;
max_N : maximum number of neighborhood cycles;
1:　　Initialize $I_n$: *% using a genetic algorithm to generate an initial feasible solution*
2:　　*gen*: = 0;
3:　　**while** *gen < MAX_gen* **do**
4:　　　　evaluate $I_n$;
5:　　　　Select $I_{n+1}$ from $I_n$; *% elitist preservation & roulette wheel selection*
6:　　　　Crossover;
7:　　　　Mutation;
8:　　　　**for** $i \leftarrow 1$ **to** *pop_size* **do**
9:　　　　　　**for** $j \leftarrow 1$ **to** *MAX_gen* **do**
10:　　　　　　　　Individual $I_{n+1}$　disturbed from the first neighborhood solution $N_1$;
11:　　　　　　　　**if** $F_{I_{n+1}} \geq F_{I_n}$ **then**
12:　　　　　　　　　　**Break**
13:　　　　　　　　**else**
14:　　　　　　　　　　iter = iter + 1;
15:　　　　　　　　**end if**
16:　　　　　　　　until (*iter = max_N*);
17:　　　　　　　　The individual $I_{n+1}$ continues to be disturbed by the next neighborhood solution $N_l$
18:　　　　　　　　Repeat
19:　　　　　　　　until (*iter = max_N*);
20:　　　　　　**end for**
21:　　　　**end for**
22:　　　　*gen = gen + 1*
23:　　**end while**
24:　Best solution

---

### 4.1. Large Neighborhood Search

For the path construction with consideration AGV type, the loading point of the least increase in cost is added to the relevant path while satisfying all constraints. If removal and insertion operators violate the constraint of time windows or capacities, LNS will evaluate the relatedness weight value of the insertion point. The removal and insertion processes will continue until no more loading points can be added into the path. The AGV type of the least cost is selected. Then the algorithm performs the next iteration process until all loading points are traversed.

While in the removal and insertion method, some parts of the flexible solution sets are entirely ruined and recreated to search a new feasible solution. This process may be to provide a better mechanism for escaping from local optima solution created by GA.

### 4.1.1. Removal Operators

The removal method in this study refers to Shaw removal (SR). The general mind behind the SR operator is to remove a set of points, as it can be expected that it is reasonably easy to change similar loading points and create better solutions. The similarity between two points $i$ and $j$ is defined by the relatedness weight $\omega(i, j)$. This includes two terms: a distance term $R'_{ij}$, a relation term $N'_{ij}$. We use the distance $d_{ij}$ between the point $i$ and $j$ to define the relatedness $R'_{ij}$ between the two points. The lower $R'_{ij}$ is, the greater relationship between these two points. Same reason as $N'_{ij}$, the relation between two points $i$ and $j$ on the same transport (delivery) path is greater than in different paths. Then $\omega(i, j)$ is defined as follows:

$$\omega(i, j) = \frac{1}{R'_{ij} + N'_{ij}} \tag{23}$$

where $R'_{ij}$ is calculated by $R'_{ij} = \frac{d_{ij}}{\max d_{ij}}$, $R'_{ij}$ is range from the interval $[0, 1]$. The value is 1 when $i$ and $j$ are on the same path, otherwise the value is 0.

### 4.1.2. Farthest Insertion Heuristic Algorithm

Suppose that $\Delta ir$ is the cost function value incurred by inserting loading point $i$ into path $r$ to the position that the increases of cost function value is least. Additionally, delimit $\cos t_i$ be the cost of inserting point $i$ at the minimum cost position. In each iteration, the farthest insertion heuristic algorithm selects point $i$ having the minimum global cost, then inserts it into the minimum global cost (i.e., $\cos t_i = \min_{i \in V} \{\Delta ir\}$), and inserts it into its minimum cost position. After one point has been inserted into its best position, $\cos t_i$ is calculated again until all removed points are inserted into the partial solution [24].

### 4.2. Chromosome Representation and the Method of Encoding

Considering the uncertainty of HFGVRPTW, including the distribution center in the coding will complicate crossover and mutation operations. In this research, sequential coding is used to generate an initial feasible solution. Besides that, the arrangement of loading points can be directly used as a chromosome, and the gene length is the number of the loading point. In the decoding process, based on the time window and capacity constraint, the loading points are assigned to the vehicles of different types. The single-vehicle loading rate is calculated, and the highest loading rate is selected for this path transportation.

The sequence coding of chromosome is shown in Figure 4.

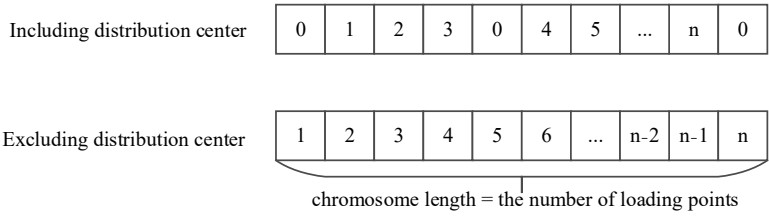

**Figure 4.** Sequence coding of chromosome.

### 4.3. Population Selection

The appropriate population size is important for the convergence of the genetic algorithm. If the population scale is too small, it is easy to converge the genetic algorithm on the local optimal solution. On the contrary, if the population size is too large, the computing speed of the genetic algorithm will be reduced. The size of the population is related to the total number of loading points $N$, and the appropriate population size should be controlled between $4N$ and $6N$ [34].

Proportionate selection is still regarded as the best selection scheme for evolutionary algorithm implementations. Using the selection scheme, the probability of an individual solution to be selected depends on its fitness value. The probability $p_j^s$ of selecting an individual solution $j$ with a fitness value $f_j$ in a population of $k$ individuals is given in the following relation in Equation (24).

$$p_j^s = \frac{f_j}{\sum\limits_{n=1}^{k} f_n} \tag{24}$$

Furthermore, considering that the traditional roulette wheel method of selecting operators cannot guarantee the uniform distribution of the next generation's solutions. In this case, we proposed a selection method called all over roulette wheel selection, to avoid the impact on the subsequent calculations of the algorithm. We set generation gap (GGAP) as the proportion of purchasing agents for the population size difference between offspring and parent. At the same time, the number of sample needles equals to the number of individuals required by the offspring. With each all over roulette wheel selection, the entire offspring can be flittered according to the fitness value, and the selection efficiency of offspring can be improved on the premise of ensuring the uniform distribution of selection results. This selection pattern is shown as in Figure 5.

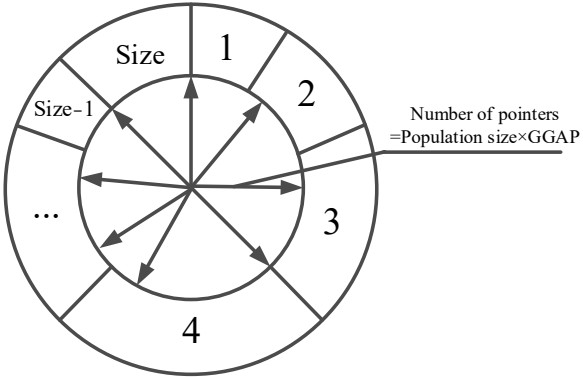

**Figure 5.** All over roulette wheel selection.

*4.4. Crossover Operation*

Traditionally, crossover refers to the process in which two chromosomes exchange some genes with each other in a certain way to form new individuals. After crossover operation, a new generation is produced, and it inherits the father's basic characteristics. The traditional two-point crossover method is that the selected gene sequences in the two chromosomes are exchanged but their positions remain unchanged. This method may cause the failure of crossover operations or increase redundant paths, eventually cause the AGV transport path to increase and affect the final optimization effective.

In this study, an improved two-pointed crossover method is selected as the operator crossover method. The steps are to randomly generate two points and swap the gene sequences at the crossover point of the two chromosomes and put them at the front of the other chromosome. Finally, in the original individual genes, the genes with the same sequence as the cross-interchangeable gene sequence are deleted sequentially. Figure 6 shows the process of crossover operation.

*4.5. Mutation Operation*

Mutation is to exchange genes within the same chromosome, resulting in a new individual. Mutation can determine the local search ability of the genetic algorithm, maintain the diversity of the population, prevent premature convergence of GA.

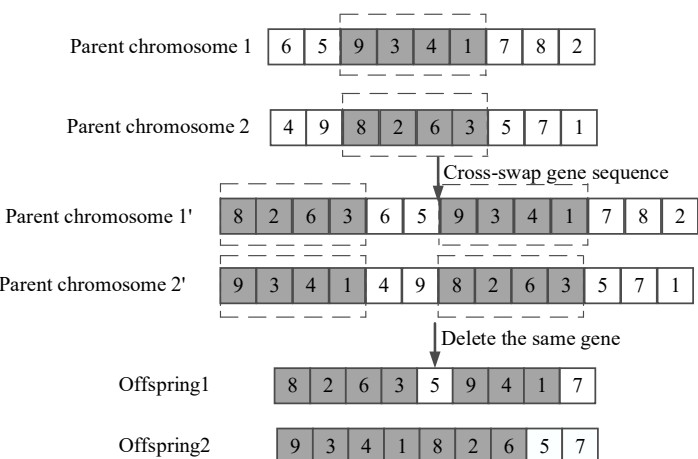

**Figure 6.** Schematic diagram of the improved two-point crossover operation process.

This paper adopts the exchanging mutation method. The idea is using the strategy of dynamic mutation, the formula for solving the mutation probability in Equation (25)

$$p_m = p_{\max} - \frac{(p_{\max} - p_{\min})}{Max\_gen} \times gen \tag{25}$$

Among them, $p_m$ represents the current mutation probability. It adaptively changes the value according to the iteration number of the algorithm. The maximum and minimum values of the mutation operation during the iteration are represented by $p_{\max}$ and $p_{\min}$, respectively. Variable *gen* and *Max_gen* indicate iterations of the algorithm currently and initially set the algorithm maximum number of iterations, respectively. The flow chart of GA-LNS algorithm structure can be seen in Figure 7.

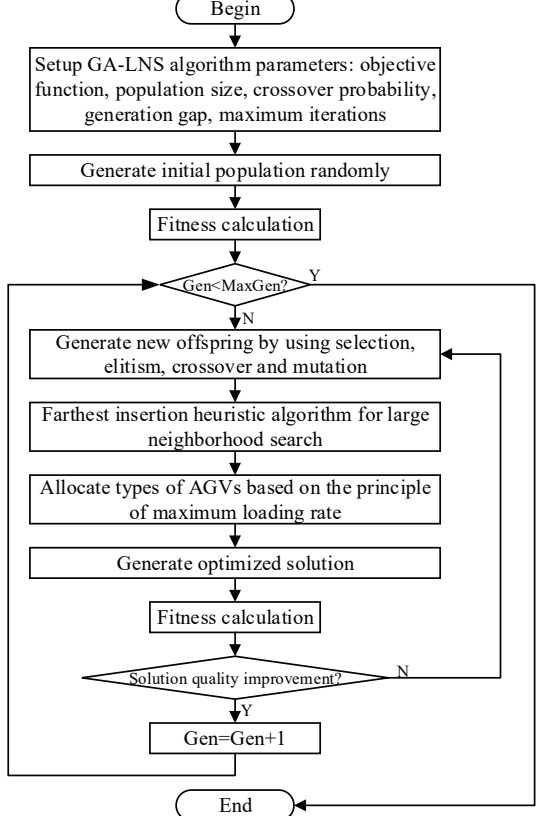

**Figure 7.** The flow-chart of GA-LNS.

The characteristics of the GA-LNS in this study are summarized as follows:

- The mutation rate is adaptively adjusted through iterative process of GA-LNS;
- The GA-LNS algorithm can not only maintain the optimization ability of large-scale neighborhood searches when solving VRP and variant problems, but also overcome the weakness of inefficiency and calculation time-consuming of traditional GA.

Thus GA-LNS is an algorithm with an excellent ability of global optimization. It can maintain the population diversity and prevent a fall into a local optimum in the calculation process.

## 5. Experimental Studies

The presented algorithm is coded in MATLAB language and was tested on a PC with Intel Core (TM) i7-8700 3.20 GHz CPU, 16 GB RAM, and Windows 10 OS. Since the HFGVRPTW is a relatively new problem and does not have a publicly standard instance [14], this type of problem can be regarded as an important extension of the classic VRPTW but divided into various categories by vehicle capacities. Thus, the widely used Solomon instance for classic VRPTW development is also applied to test the performance of GA-LNS algorithm [37]. This computational experiment is divided into two phases of performance analysis, both in terms of effectiveness and energy-saving effect. In the effectiveness analysis, we use GA-LNS to solve the benchmark instances of different customers' scales to investigate the efficiency and performance of our proposed algorithm. In the energy-saving effect, we use an experiment on real FMS to measure the energy-saving performance of the proposed loading method.

### 5.1. Instances and Results Comparison

The Solomon's benchmark instance was proposed in 1987. It composed of three different distribution pattern of customers locations which is divided into six categories [37], with clustered (C1, C2), randomly located (R1, R2), or semi-clustered (RC1, RC2). According to reference [38], a selected Solomon's benchmark instance set can be used to test the performance of an algorithm.

In Solomon's benchmark instances, the customer geographical coordinates, demands and service times are the same. The difference between the groups of instances indexed by 1 and those indexed by 2 stand the intervals density of time windows. Time window density refers to the percentage of customers with time windows. Furthermore, in the instance sets of Solomon, 01/05/09, 02/06/10, 03/07/11, and 04/08/12 respond to the instances with 100%, 75%, 50%, and 25% time windows, respectively [7]. According to the scale of the test instances, Solomon's benchmark instances can be divided into three problem scales, namely, the small-scale instances with 25 customers, the medium-scale with 50 customers, and the large-scale instances with 100 customers [39].

In this study, in order to illustrate the effectiveness of GA-LNS in solving VRPTW, we selected six sets of instances with different time window densities under three different customers scales. For each instance, we have run the program for 10 times, then a statistics analysis is undertaken for the computing results.

Tables 3–5 present the results of three different customer scales. To illustrate the advantages and disadvantages of the GA-LNS, we analyze the experimental mathematical results obtained by GA-LNS by comparing with the best results published and other intelligent algorithms for the large-scale of customers is shown in Tables 6 and 7.

As shown in Tables 3 and 4, the following observations can be obtained. The solutions obtained by GA-LNS are very similar with the published best results, considering either the total of traveled distance or vehicle fleet size. As we can be seen, in Table 3, the proposed algorithm performs extremely well on the Solomon instance. Besides that, improves or achieves the best-known solution of the hybrid GA to 17 of 24 instances. Moreover, it is about 29.2% better than the hybrid GA. For these medium-scale instances, the gap between the best-known solution and the optimal solution varies from 0.03% to 1.93% while the

average gap is 0.32%. To conclude, the proposed algorithm can achieve a competitive solution and the effectiveness is proven through small and medium-scale instances.

**Table 3.** The experimental results for the small-scale Solomon's instances with VRPTW model.

| NO. | Best-Known Solution [37] | Hybrid GA [39] | GA-LNS | | | | | |
|---|---|---|---|---|---|---|---|---|
| | NV/TD | avg. NV/best. TD | avg. NV | avg. TD | best. TD | worst. TD | TD. gap% | best. NV/best. TD |
| C101 | 3/191.3 | 3.0/191.81 | 3.0 | 191.40 | 191.40 | 191.40 | 0.05 | 3/191.40 |
| C102 | 3/190.3 | 3.0/190.74 | 3.0 | 190.33 | 190.33 | 190.33 | 0.02 | 3/190.33 |
| C103 | 3/190.3 | 3.0/190.74 | 3.0 | 189.71 | 189.71 | 189.71 | −0.31 | 3/189.71 |
| C104 | 3/186.9 | 3.0/187.45 | 3.0 | 189.71 | 189.71 | 189.71 | 1.50 | 3/189.71 |
| C205 | 2/214.7 | 2.0/215.54 | 2.0 | 215.54 | 215.54 | 215.54 | 0.39 | 2/215.54 |
| C206 | 3/214.7 | 2.0/215.54 | 2.0 | 215.54 | 215.54 | 215.54 | 0.39 | 2/215.54 |
| C207 | 2/214.5 | 2.0/215.34 | 2.0 | 215.34 | 215.34 | 215.34 | 0.39 | 2/215.34 |
| C208 | 2/214.5 | - | 2.0 | 215.37 | 215.37 | 215.37 | 0.41 | 2/215.37 |
| R105 | 6/530.5 | 6.0/531.54 | 6.0 | 538.25 | 542.26 | 537.25 | 1.27 | 6/537.25 |
| R106 | 3/465.4 | 5.0/466.48 | 5.0 | 466.48 | 466.48 | 466.48 | 0.23 | 5/466.48 |
| R107 | 4/424.3 | 4.6/425.27 | 4.0 | 425.27 | 425.27 | 425.27 | 0.23 | 4/425.27 |
| R108 | 4/397.3 | 4.0/398.29 | 4.0 | 411.12 | 409.20 | 413.23 | 3.00 | 4/409.20 |
| R205 | 3/393.0 | 2.8/395.82 | 3.0 | 394.06 | 394.06 | 394.06 | 0.27 | 3/394.06 |
| R206 | 3/374.4 | 1.8/379.85 | 3.0 | 375.48 | 375.48 | 375.48 | 0.29 | 3/375.48 |
| R207 | 3/361.6 | 1.8/369.21 | 3.0 | 362.63 | 362.63 | 362.63 | 0.28 | 3/362.63 |
| R208 | 1/328.2 | - | 1.0 | 329.33 | 329.33 | 329.33 | 0.34 | 1/329.33 |
| RC101 | 4/461.1 | 4.0/462.61 | 4.0 | 463.60 | 463.60 | 463.60 | 0.54 | 4/463.60 |
| RC102 | 3/351.8 | 3.0/352.74 | 3.0 | 352.74 | 352.74 | 352.74 | 0.27 | 3/352.74 |
| RC103 | 3/332.8 | 3.0/333.92 | 3.0 | 333.92 | 333.92 | 333.92 | 0.34 | 3/333.92 |
| RC104 | 3/306.6 | 3.0/308.39 | 3.0 | 308.39 | 308.39 | 308.39 | 0.58 | 3/308.39 |
| RC205 | 3/338.0 | 3.0/338.93 | 3.0 | 338.93 | 338.93 | 338.93 | 0.28 | 3/338.93 |
| RC206 | 3/324.0 | 2.4/326.14 | 3.0 | 325.10 | 325.10 | 325.10 | 0.34 | 3/325.10 |
| RC207 | 3/298.3 | 2.6/298.95 | 3.0 | 298.95 | 298.95 | 298.95 | 0.22 | 3/298.95 |
| RC208 | 2/269.1 | - | 2.0 | 269.57 | 269.57 | 269.57 | 0.17 | 2/269.57 |

NO.: the ID of the instance; NV: the number of the used vehicle; TD: total travelling distance; avg. NV: the average number of the used vehicle in the 10 running times; best. TD: the smallest objective value obtained in the 10 running times; avg. TD: the average smallest objective value obtained in the 10 running times; worst. TD: the biggest objective value obtained in the 10 running times; TD. gap%: (best. TD − the distance of best-known solution)/the distance of best-known solution × 100%.

**Table 4.** The experimental results for the medium-scale of Solomon's instances with VRPTW model.

| NO. | Best-Known Solution [37] | GA-LNS | | | | | |
|---|---|---|---|---|---|---|---|
| | NV/TD | avg. NV | avg. TD | best. TD | worst. TD | TD. gap% | best. NV/best. TD |
| C101 | 5/362.4 | 5.0 | 362.81 | 362.81 | 362.81 | 0.11 | 5/362.81 |
| C102 | 5/361.4 | 5.0 | 361.74 | 361.74 | 361.74 | 0.09 | 5/361.74 |
| C103 | 5/361.4 | 5.0 | 366.33 | 361.64 | 385.10 | 0.07 | 5/361.64 |
| C104 | 5/358.0 | 5.0 | 362.60 | 358.17 | 378.43 | 0.05 | 5/358.17 |
| C205 | 3/359.8 | 3.0 | 359.50 | 356.14 | 360.14 | 0.09 | 3/360.14 |
| C206 | 3/359.8 | 3.0 | 359.92 | 359.92 | 359.92 | 0.03 | 3/359.92 |
| C207 | 3/359.6 | 3.0 | 360.21 | 360.21 | 360.21 | 0.17 | 3/360.21 |
| C208 | 2/350.5 | 2.0 | 351.71 | 351.71 | 351.71 | 0.35 | 2/351.71 |
| R105 | 9/899.3 | 9.0 | 910.10 | 900.85 | 915.35 | 0.28 | 9/900.85 |
| R106 | 5/793 | 5.0 | 799.39 | 795.23 | 809.90 | 0.11 | 5/795.23 |
| R107 | 7/711.1 | 7.0 | 711.85 | 711.85 | 711.85 | 1.01 | 7/711.85 |
| R108 | 6/617.7 | 6.0 | 628.90 | 623.85 | 634.94 | 0.34 | 6/623.94 |
| R205 | 4/690.1 | 4.2 | 694.97 | 692.46 | 694.98 | 0.33 | 4/692.46 |
| R210 | 4/645.6 | 3.4 | 647.71 | 647.71 | 647.71 | 0.52 | 3/647.71 |
| R211 | 3/535.5 | 3.0 | 538.44 | 538.29 | 538.71 | 0.19 | 3/538.30 |
| R204 | 2/506.4 | 2.0 | 507.34 | 507.34 | 507.34 | 0.54 | 2/507.34 |
| RC101 | 8/944 | 8.2 | 957.31 | 949.10 | 967.16 | 0.37 | 8/949.10 |
| RC102 | 7/822.5 | 7.0 | 831.77 | 825.58 | 834.80 | 0.17 | 7/825.58 |
| RC103 | 6/710.9 | 6.4 | 706.25 | 693.05 | 724.60 | 1.93 | 6/724.60 |
| RC104 | 5/545.8 | 5.0 | 546.57 | 546.25 | 547.87 | 0.08 | 5/546.25 |

**Table 4.** *Cont.*

| NO. | Best-Known Solution [37] | GA-LNS | | | | | |
|---|---|---|---|---|---|---|---|
| | NV/TD | avg. NV | avg. TD | best. TD | worst. TD | TD. gap% | best. NV/best. TD |
| RC205 | 5/630.2 | 5.0 | 631.38 | 631.07 | 632.63 | 0.39 | 5/632.63 |
| RC206 | 5/610.0 | 5.0 | 610.95 | 610.77 | 611.68 | 0.13 | 5/610.77 |
| RC207 | 4/558.6 | 4.2 | 559.30 | 559.00 | 560.51 | 0.07 | 4/559.00 |
| RC204 | 3/444.2 | 3.0 | 444.97 | 444.97 | 444.97 | 0.17 | 3/444.97 |

**Table 5.** The experimental results for the large-scale of Solomon's instances with VRPTW model.

| NO. | Best-Known Solution [37] | GA-LNS | | | | | |
|---|---|---|---|---|---|---|---|
| | NV/TD | avg. NV | avg. TD | best. TD | worst. TD | TD. gap% | best. NV/best. TD |
| C101 | 10/828.94 | 10.0 | 829.62 | 829.62 | 829.62 | 0.08 | 10/829.62 |
| C102 | 10/828.94 | 10.2 | 871.98 | 829.44 | 901.15 | 0.06 | 10/829.44 |
| C103 | 10/828.06 | 10.4 | 843.50 | 829.32 | 864.77 | 0.15 | 10/829.32 |
| C104 | 10/824.78 | 10.0 | 856.39 | 827.42 | 899.85 | 0.32 | 10/827.42 |
| C205 | 3/588.88 | 3.0 | 588.95 | 588.95 | 588.95 | 0.01 | 3/588.95 |
| C206 | 3/588.49 | 3.0 | 588.56 | 588.56 | 588.56 | 0.01 | 3/588.56 |
| C207 | 3/588.29 | 3.0 | 589.96 | 589.96 | 589.96 | 0.28 | 3/589.96 |
| C208 | 3/588.32 | 3.0 | 588.34 | 588.34 | 588.34 | 0.00 | 3/588.34 |
| R105 | 14/1377.11 | 15.0 | 1446.88 | 1379.02 | 1494.06 | 0.14 | 14/1379.02 |
| R106 | 12/1252.03 | 12.4 | 1257.85 | 1256.73 | 1259.53 | 0.38 | 12/1256.73 |
| R107 | 10/1104.66 | 10.2 | 1111.25 | 1105.43 | 1127.03 | 0.07 | 10/1105.43 |
| R108 | 9/960.88 | 9.6 | 969.60 | 966.61 | 974.08 | 1.37 | 9/974.08 |
| R205 | 3/994.43 | 3.4 | 997.76 | 997.20 | 999.14 | 0.28 | 3/997.20 |
| R206 | 3/906.14 | 3.8 | 914.05 | 913.15 | 914.97 | 0.87 | 3/914.02 |
| R207 | 2/890.61 | 3.3 | 829.25 | 808.00 | 846.40 | −9.3 | 3/808.00 |
| R208 | 2/726.82 | 3.0 | 734.58 | 729.31 | 758.2956 | −0.07 | 2/726.32 |
| RC101 | 14/1696.95 | 16.4 | 1766.89 | 1728.57 | 1774.19 | 3.41 | 16/1754.85 |
| RC102 | 12/1554.75 | 14.8 | 1569.32 | 1556.70 | 1570.22 | 0.13 | 14/1556.70 |
| RC103 | 11/1261.67 | 10.0 | 1401.54 | 1378.23 | 1416.99 | 9.24 | 12/1378.23 |
| RC104 | 10/1135.48 | 11.0 | 1255.43 | 1210.58 | 1302.60 | 6.61 | 11/1210.58 |
| RC205 | 4/1297.65 | 5.6 | 1193.14 | 1173.71 | 1221.3216 | −9.55 | 5/1173.71 |
| RC206 | 3/1146.32 | 4.4 | 1097.70 | 1081.84 | 1115.87 | −5.62 | 4/1081.84 |
| RC207 | 3/1061.41 | 4.7 | 1008.31 | 1003.09 | 1018.44 | −5.49 | 5/1003.09 |
| RC208 | 3/828.14 | 4.2 | 818.44 | 807.94 | 834.89 | −2.44 | 4/807.94 |

**Table 6.** The other intelligent algorithms experimental results for the large-scale Solomon's instances.

| NO. | Saving Algorithm [40] | LNS [40] | p-SA [41] | GA [42] | ACO [43] | SA-LNS [44] | HACO [43] |
|---|---|---|---|---|---|---|---|
| | NV/TD | NV/TD | NV/TD | NV/TD | NV/TD | NV/TD | NV/TD |
| C101 | 12/930.12 | 10/828.94 | 11/992.88 | 10/827.3 | 13/1299.02 | 10/828.94 | 13/1262.53 |
| C102 | 13/1048.17 | 10/828.94 | 10/955.31 | 10/827.3 | 13/1538.73 | 10/828.94 | 13/1693.11 |
| C103 | 12/997.49 | 10/875.25 | 10/958.66 | 10/826.3 | 11/1669.38 | 10/828.07 | 11/1530.39 |
| C104 | 12/964.77 | 10/883.67 | 10/944.73 | 10/822.9 | 10/1278.60 | 10/824.78 | 10/1307.09 |
| C205 | 5/729.12 | 3/588.88 | 3/588.88 | 3/586.4 | 4/674.47 | 3/588.88 | 4/669.25 |
| C206 | 5/727.62 | 3/588.49 | 3/588.49 | 3/586.0 | 4/776.75 | 3/588.49 | 4/753.34 |
| C207 | 5/745.11 | 3/588.29 | 3/588.29 | 3/585.8 | 4/756.17 | 3/588.29 | 4/715.13 |
| C208 | 5/711.55 | 3/592.30 | 3/588.32 | 3/585.8 | 4/773.63 | 4/588.32 | 4/720.31 |
| R105 | 22/1628.90 | 14/1409.68 | 14/1399.81 | 13/1355.3 | 19/1966.78 | 14/1377.11 | 19/1870.37 |
| R106 | 20/1497.98 | 12/1286.92 | 12/1275.69 | 13/1234.6 | 16/1859.20 | 12/1252.03 | 16/1843.72 |
| R107 | 19/1398.83 | 11/1171.87 | 11/1082.92 | 11/1067.3 | 13/1647.18 | 10/1104.66 | 13/1630.95 |
| R108 | 13/1094.07 | 9/1019.33 | 10/962.48 | 10/946.2 | 12/1317.30 | 9/960.88 | 12/1285.82 |

**Table 6.** *Cont.*

| NO. | Saving Algorithm [40] | LNS [40] | p-SA [41] | GA [42] | ACO [43] | SA-LNS [44] | HACO [43] |
|-----|------------------------|----------|-----------|---------|----------|-------------|-----------|
| | NV/TD | NV/TD | NV/TD | NV/TD | NV/TD | NV/TD | NV/TD |
| R205 | 12/1146.62 | 3/1002.62 | 3/1046.06 | 5/972.5 | 4/1495.27 | 3/994.42 | 4/1470.54 |
| R206 | 12/1103.85 | 3/950.32 | 3/959.94 | 5/934.4 | 3/1470.71 | 3/914.63 | 3/1455.13 |
| R207 | 8/943.53 | 2/926.12 | 2/899.82 | 4/860.1 | 3/1339.77 | 2/726.82 | 3/1333.29 |
| R208 | 6/822.44 | 2/726.82 | 2/739.06 | 4/744.1 | 3/1088.24 | 3/1088.24 | 3/1044.05 |
| RC101 | 25/2117.08 | 14/1782.30 | 15/1659.59 | 15/1623.5 | 21/2444.21 | 14/1696.95 | 21/2350.94 |
| RC102 | 21/1830.68 | 12/1592.62 | 13/1522.76 | 14/1458.2 | 16/2142.05 | 12/1554.75 | 16/2132.71 |
| RC103 | 16/1549.89 | 12/1275.43 | 11/1344.62 | 12/272.5 | 13/1802.48 | 11/1261.67 | 13/1791.28 |
| RC104 | 13/1272.40 | 10/1196.30 | 10/1268.30 | 10/1137.6 | 12/1655.40 | 10/1135.48 | 12/1643.07 |
| RC205 | 16/1572.14 | 4/1348.73 | 4/1371.08 | 7/1154.0 | 5/2138.76 | 4/1297.65 | 5/2096.55 |
| RC206 | 13/1444.01 | 3/1197.63 | 3/1166.88 | 6/1080.4 | 4/1729.86 | 3/1146.32 | 4/1748.73 |
| RC207 | 11/1256.69 | 3/1112.49 | 3/1089.85 | 7/1005.6 | 4/1656.32 | 3/1061.14 | 4/1618.84 |
| RC208 | 5/828.849 | 3/829.69 | 3/862.89 | 5/820.5 | 3/1357.87 | 3/828.14 | 3/1306.25 |

**Table 7.** Comparison between the mean numbers of vehicle number and total travelling distance.

| Algorithm | | C1 | C2 | R1 | R2 | RC1 | RC2 |
|-----------|------|-------|-------|---------|---------|---------|---------|
| Published Best [37] | NV | 10.00 | 3.00 | 11.25 | 2.50 | 11.75 | 3.25 |
| | TD | 827.68 | 588.50 | 1173.67 | 879.50 | 1412.21 | 1083.38 |
| GA-LNS | NV | 10.00 | 3.00 | 11.25 | 2.75 | 13.25 | 4.50 |
| | TD | 828.95 | 588.95 | 1178.82 | 861.39 | 1475.09 | 1016.64 |
| | TD% | 0.15 | 0.08 | 0.44 | −2.06 | 4.45 | −6.16 |
| Saving Algorithm [40] | NV | 12.25 | 5.00 | 18.50 | 9.50 | 18.75 | 12.00 |
| | TD | 985.14 | 728.35 | 1404.95 | 1004.11 | 1692.51 | 1275.42 |
| | TD% | 19.02 | 23.76 | 19.70 | 14.17 | 19.85 | 17.73 |
| LNS [40] | NV | 10.00 | 3.00 | 11.50 | 2.50 | 12.00 | 3.25 |
| | TD | 854.20 | 589.49 | 1221.95 | 901.47 | 1461.66 | 1122.14 |
| | TD% | 3.20 | 0.17 | 4.11 | 2.50 | 3.50 | 3.58 |
| p-SA [41] | NV | 10.25 | 3.00 | 11.75 | 2.50 | 12.25 | 3.25 |
| | TD | 962.90 | 588.50 | 1180.23 | 911.22 | 1448.82 | 1122.68 |
| | TD% | 16.33 | 0.00 | 0.44 | 3.46 | 3.50 | 3.58 |
| GA [42] | NV | 10.00 | 3.00 | 11.75 | 4.50 | 12.75 | 6.25 |
| | TD | 825.95 | 586.00 | 1150.85 | 877.78 | 1122.95 | 1015.13 |
| | TD% | −0.21 | −0.42 | −1.94 | −0.196 | −20.48 | −6.30 |
| ACO [43] | NV | 11.75 | 4.00 | 15.00 | 3.25 | 15.50 | 4.00 |
| | TD | 1446.43 | 745.26 | 1697.62 | 1348.50 | 2011.04 | 1720.70 |
| | TD% | 74.76 | 26.64 | 44.64 | 53.33 | 42.40 | 58.83 |
| SA-LNS [44] | NV | 10.00 | 3.25 | 11.25 | 2.75 | 11.75 | 3.25 |
| | TD | 827.68 | 588.50 | 1173.67 | 931.03 | 1412.21 | 1083.31 |
| | TD% | 0.00 | 0.00 | 0.00 | 5.86 | 0.00 | −0.01 |
| HACO [43] | NV | 11.75 | 4.00 | 15.00 | 3.25 | 15.50 | 4.00 |
| | TD | 1448.28 | 714.51 | 1657.72 | 1325.75 | 1979.50 | 1692.59 |
| | TD% | 74.98 | 21.41 | 41.24 | 50.74 | 40.17 | 56.23 |

In most large-scale Solomon instances, GA-LNS can maintain a competitive optimal solution as shown in Table 5. For the categories of C and R instances, the gap between the best-known solution and the computational solution to the proposed algorithm varies from 0.00% to 9.55% while the average gap is 2.33%. Another phenomenon should be mentioned is that in large-scale RC instances, GA-LNS show more fluctuation performances than other categories of instances. The reason may be that in RC type instances, the original problem instances have conflict solution objectives. Since customer points are relatively discrete and some customers have relatively long service time windows, which means the

reducing fleet of vehicle size will increase travelling distance, and the performance of the algorithm becomes unstable. We can further the optimization performance of the algorithm by increasing the population size and evolution algebra.

In Table 6, the saving algorithm, LNS, simulated annealing algorithm (SA), GA, ant colony optimization (ACO) as well as simulated annealing algorithm based on large neighborhood search (SA-LNS) and improved ant colony algorithm (HACO) that can represent hybrid optimization algorithms were adopted, respectively. References [40–44] use the same as numerical instances this study which can used for an indirect comparison with the algorithm performance. The details of these proposed algorithms can be found in the relevant references, and the results of a comparison of these algorithms are given in Table 6 together.

Table 7 summarizes the average results from each of the Solomon data sets of the large-scale problems. Three numbers are associated with the average number of traveled vehicle (NV), average total traveled distance (TD) and the increased percentages from the published best results (TD%) for each paper. It can be seen from Table 7 that ACO algorithm has the worst solving quality among the 24 groups of instances, with a maximum gap is 74.76% and the average gap is 50.10%. The solution quality of p-SA and GA are general, with a maximum gap of 16.33% and 20.48% and average gap of 4.55% and 4.92%, respectively. The solution quality of GA-LNS is better, with a maximum gap of 6.16% and an average gap of 2.22%. In this paper, the SA-LNS solution quality is the best: the maximum gap is 5.86% and the average gap is 0.98%. Although it is worth clarifying that works [40,44] have considered the minimum number of vehicles as the first optimization objective, the other works have considered the total traveled distance as the first objective. The vehicle fleet size is fitness in terms of references [40,44]. GA-LNS reduced the vehicle number effectively, an average of five fewer vehicles compared with the saving algorithm, it is basically the same as the fleet size solved by [44]. The performance of GA-LNS is slightly lower than that in [44] in terms of traveling distance, but higher than that in [40–43]. However, the solution results of most instances confirm the viability of GA-LNS approaches for traveled distance minimization in the VRPTW and reduction of distribution cost. Figure 8 shows the GAP curve for these Solomon instances. The horizontal axis represents the number of instances in Tables 6 and 7, the vertical axis is the optimal solution result of proposed algorithm. From the Figure 8, it is easy to conclude that GA-LNS can achieve an acceptable optimization effect. Furthermore, it performs better than most of other algorithms which are used for comparison in this article. Therefore, it can be concluded that the proposed algorithms has a relatively strong robustness and optimization ability, it can provide high-quality solutions to the HFGVRPTW.

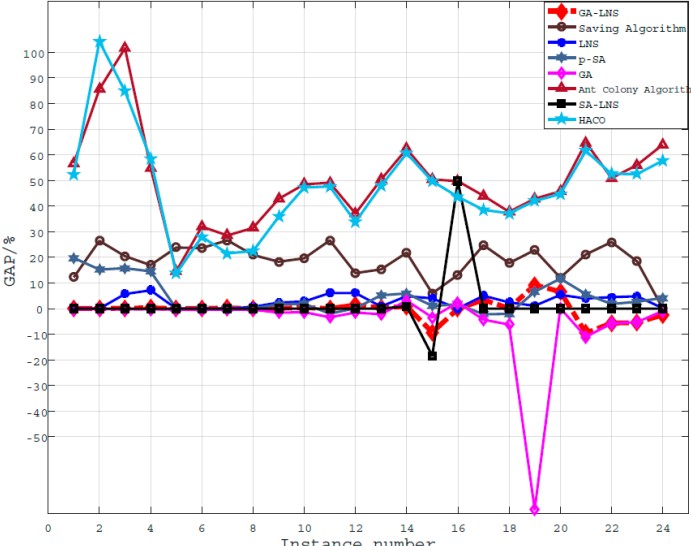

**Figure 8.** Results of all algorithms compared with the optimal solution.

### 5.2. Instance Verification

To verify the benefits of the proposed energy-saving model, explore the factors affecting optimization objectives of AGV in the process of performing production tasks, this experiment was carried out the simulation of heterogeneous multitype fleet AGVs in an actual FMS.

#### 5.2.1. Real Case Application

The FMS was mainly composed of automated workstations, automated flexible material handling system (conveyors and AGVs), dedicated dispatch control system and a buffer for storing materials, which can realize the automatic production of some high-precision parts. The workshop layout of this FMS is shown in Figure 9.

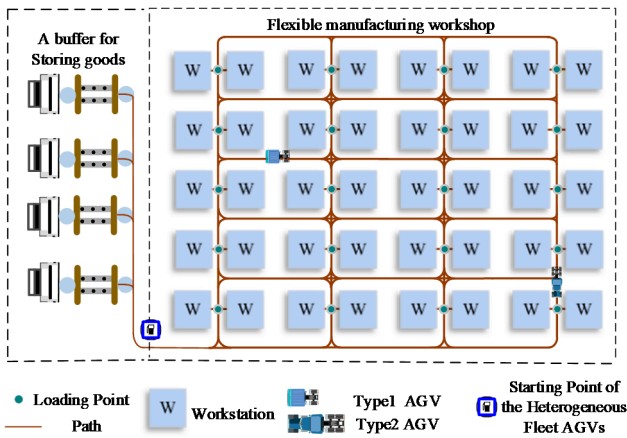

**Figure 9.** FMS workshop layout.

AGV is a flexible material handling device that is automatically steered to accomplish its assigned task inside the system. The corresponding production task is to move the materials that have been produced from the workstation to the buffer for storing materials area for storage. After the dedicated dispatch control system assigns tasks and the path planning for one AGV, the AGV sets out from the starting point, travels along the assigned path, completes the corresponding transportation task and returns to the starting point to wait for next schedule.

The FMS uses two types of AGVs to providing materials transportation. The main technical parameters of the heterogeneous fleet AGVs are listed in Table 8.

**Table 8.** The main technical parameters of the heterogeneous fleet AGVs.

| Parameter | Meaning | Value | |
|---|---|---|---|
| | | **Type 1** | **Type 2** |
| $L \times W \times H$ | External dimension | 0.8 m × 0.6 m × 0.4 m | 1.2 m × 0.6 m × 0.5 m |
| $C_m$ | Maximal load weight | 100 kg | 200 kg |
| $F_m$ | Motor rated power output | 130 N | 200 N |
| $E_{sm}$ | Standby power | 65 J | 130 J |
| $R_m$ | Turning radius | 0.60 m | 0.85 m |
| $v_r$ | Turning speed | 0.5 m/s | |
| $v_s$ | Uniform straight motion speed | 1 m/s | |
| $a_{acc}$ | Acceleration of accelerated motion | 1 m/s$^2$ | |
| $a_{dec}$ | Acceleration of decelerated motion | 1 m/s$^2$ | |
| $\eta$ | Driving motor efficiency | 0.95 | 0.90 |

The docking stations for loading and path intersections were abstracted as nodes, and each edge presents a transportation path laid by the magnetic stripe. The path attribute is defined as a bidirectional path with a single unit capacity. Therefore, there will not be any

deadlocks and conflicts between AGVs. Twenty instances were generated to validate the energy effectiveness and optimization of the proposed model and mathematical algorithm. Instances are represented in the following Table 9. The distance between two nodes $i$ and $j$ is equal to the Manhattan distance between them. In this actual system, there is no production priority between workstations and each AGV corresponds to one assigned transportation path.

**Table 9.** Time window and goods weight for each loading point.

| Loading Point ID | Coordinates (m) | $LT_n$ | $RT_n$ | Materials (kg) | Loading Time (min) |
|---|---|---|---|---|---|
| 0 | (0, 0) | 0 | $+\infty$ | - | - |
| 1 | (6.5, 4.5) | 9:10 | 9:45 | 25 | 2.0 |
| 2 | (6.5, 10.5) | 9:15 | 9:40 | 20 | 1.5 |
| 3 | (6.5, 16.5) | 10:10 | 10:30 | 15 | 1.0 |
| 4 | (6.5, 22.5) | 9:30 | 10:20 | 60 | 4.0 |
| 5 | (6.5, 28.5) | 10:25 | 11:00 | 25 | 1.5 |
| 6 | (16.5, 4.5) | 9:40 | 10:20 | 30 | 1.5 |
| 7 | (16.5, 10.5) | 10:25 | 11:40 | 70 | 4.0 |
| 8 | (16.5, 16.5) | 10:40 | 12:10 | 80 | 3.0 |
| 9 | (16.5, 22.5) | 10:30 | 11:15 | 30 | 2.0 |
| 10 | (16.5, 28.5) | 10:40 | 12:00 | 60 | 2.5 |
| 11 | (26.5, 4.5) | 9:20 | 10:00 | 45 | 3.0 |
| 12 | (26.5, 10.5) | 11:10 | 11:50 | 35 | 1.5 |
| 13 | (26.5, 16.5) | 11:30 | 11:50 | 20 | 1.0 |
| 14 | (26.5, 22.5) | 9:40 | 10:30 | 55 | 3.0 |
| 15 | (26.5, 28.5) | 9:45 | 10:40 | 70 | 4.0 |
| 16 | (36.5, 4.5) | 10:25 | 10:50 | 15 | 0.5 |
| 17 | (36.5, 10.5) | 11:00 | 12:00 | 55 | 2.0 |
| 18 | (36.5, 16.5) | 11:10 | 11:40 | 20 | 1.0 |
| 19 | (36.5, 22.5) | 10:00 | 10:30 | 20 | 1.5 |
| 20 | (36.5, 28.5) | 10:20 | 11:00 | 35 | 2.0 |

During a representative production time period, we compared the energy consumption traveled by the single type AGVs with the heterogeneous multitype AGVs solutions produced by our algorithm. For this instance, we ran the program for 10 times, then a statistics analysis was undertaken to compute results.

### 5.2.2. Numerical Analysis of Experimental Results

Tables 10 and 11 correspond to the result of the homogeneous multitype fleet using GA-LNS algorithm to solve the actual instance and Table 12 represents the path information loaded by heterogeneous multitype AGVs fleet.

**Table 10.** Feasible paths for using type 1 AGV fleet transportation.

| No. | Path | Optimization Objective | | $n_T$ | Loading Rate/% |
|---|---|---|---|---|---|
| | | D [m] | E [kJ] | | |
| 1 | 0-1-13-18-19-0 | 118.80 | 17.40 | 6 | 85.00 |
| 2 | 0-4-0 | 54.40 | 7.83 | 2 | 60.00 |
| 3 | 0-6-16-17-0 | 95.80 | 13.88 | 4 | 100.00 |
| 4 | 0-7-0 | 62.40 | 8.93 | 2 | 70.00 |
| 5 | 0-8-0 | 171.90 | 24.49 | 2 | 80.00 |
| 6 | 0-11-12-0 | 70.40 | 10.02 | 2 | 80.00 |
| 7 | 0-14-0 | 94.40 | 13.31 | 2 | 55.00 |
| 8 | 0-15-0 | 106.40 | 14.95 | 2 | 70.00 |
| 9 | 0-20-10-0 | 133.60 | 19.05 | 4 | 95.00 |
| 10 | 0-2-3-5-9-0 | 90.60 | 13.16 | 4 | 90.00 |

**Table 11.** Feasible paths for using type 2 AGV fleet transportation.

| No. | Path | Optimization Objective | | $n_T$ | Loading Rate/% |
| | | D [m] | E [kJ] | | |
|---|---|---|---|---|---|
| 1 | 0-4-3-5-0 | 80.35 | 18.59 | 2 | 50.00 |
| 2 | 0-11-16-17-7-0 | 93.85 | 25.16 | 6 | 92.50 |
| 3 | 0-14-9-8-0 | 98.60 | 23.38 | 4 | 82.50 |
| 4 | 0-15-10-0 | 110.60 | 26.05 | 4 | 65.00 |
| 5 | 0-20-19-18-13-12-0 | 132.30 | 30.86 | 4 | 65.00 |
| 6 | 0-2-6-1-0 | 55.10 | 14.47 | 6 | 37.50 |

**Table 12.** Feasible paths for using heterogeneous multitype AGV fleet transportation.

| No. | AGV Type | Path | Optimization Objective | | $n_T$ | Loading Rate/% |
| | | | D [m] | E [kJ] | | |
|---|---|---|---|---|---|---|
| 1 | 2 | 0-4-5-10-0 | 83.10 | 19.94 | 4 | 72.50 |
| 2 | 1 | 0-6-9-12-0 | 100.20 | 14.48 | 4 | 95.00 |
| 3 | 2 | 0-11-16-17-7-0 | 100.25 | 24.49 | 6 | 92.50 |
| 4 | 2 | 0-14-20-15-0 | 129.40 | 30.69 | 6 | 80.00 |
| 5 | 1 | 0-1-13-18-19-0 | 115.90 | 17.00 | 6 | 85.00 |
| 6 | 2 | 0-2-3-8-0 | 66.60 | 16.28 | 4 | 57.50 |

Tables 10 and 11 show that the distribution center sends 10 type 1 AGVs or six type 2 AGVs to serve all loading points. For a homogeneous vehicle fleet which is using the type 1 AGVs, the total traverse distance and energy consumption are 998.7 m and 143.02 kJ. Besides, for the homogeneous vehicle fleet, which is composed of type 2 AGVs, the total traverse distance and energy consumption are 570.80 m and 138.51 kJ, respectively.

In Table 12, the different types of AGVs are compared. We compared the heterogeneous AGVs fleet with the type 1 and type 2. Table 13 shows the results that the benefit of using the heterogeneous vehicle fleet. As shown as Table 13, compared with single use type 1, the heterogeneous fleet AGVs in terms of $D_{total}$ can be reduced by 40.38% from 998.7 to 595.45, $E_{total}$ reduced by 13.89%, average fleet loading rate increased from 78.50% to 80.42%.

**Table 13.** Comparison of objective corresponding to the optimal paths under three conditions.

| Path Information | Transport AGV Type | | |
| | Type 1 | Type 2 | Heterogeneous Fleet |
|---|---|---|---|
| $D_{total}$ (m) | 998.70 | 570.80 | 595.45 |
| $E_{total}$ (kJ) | 143.02 | 138.51 | 123.15 |
| Fleet size | 10 | 6 | 6 |
| Avg. Loading rate/% | 78.50 | 65.42 | 80.42 |

This comparison clearly shows that, at least for the actual instance considered in this paper, heterogeneous fleet distribution strategy can significantly reduce $E_{total}$ and $D_{total}$ under the premise of ensuring average loading rate.

In addition, it is worth noting that the fleet size of using type 1 transportation is 10, the fleet size of using heterogeneous fleet transportation is 6. Inside the workshop of FMS, in the case of excessive traffic volume, it will undoubtedly cause frequent conflicts and deadlocks between AGVs. While causing a decrease in production efficiency, it will have negative impact on the production plan. Therefore, the use of heterogeneous multitype AGV fleets in FMS has a positive effect on reducing the traffic flow inside the workshop.

When using type 2 AGVs to execute the loading tasks, the $E_{total}$ of heterogeneous fleets reduced from 138.51 to 123.15, the average loading rate of the fleet increased significantly, whereas the traveled distance increased by only 4.32% from 570.8 to 595.45. In general, the experiments results demonstrate that it is necessary to investigate the features and the

benefits of HFGVRPTW in more details and to analyze carefully the tradeoff between the solution of heterogeneous fleet of AGVs. Additionally, for the homogeneous fleet consisting of type 2, the decreased number of AGVs leading to single AGV task chain is too long. As a result, it is difficult to respond to loading tasks in FMS in a timely manner, and the AGV loading rate is also maintained at a low level while affecting the punctual rate of tasks in the entire workshop. In the heterogeneous multitype AGV fleet which are not full load in the instances of same time windows. For example, the first AGV in Table 11 has a loading rate of only 50% when returning to the starting point. After loading and transporting with a heterogeneous fleet, loading point 3 is replaced with point 10 of the large load. This method can increase the average load rate of the vehicle fleet.

To sum up, through the application and analysis of instances, it can be shown that the proposed GA-LNS effective and robustness is well. In FMS, introducing heterogeneous fleet of AGVs with different capacities and energy consumption opening a tremendous potential for energy-saving.

## 6. Conclusions and Future Research Work

In this paper, the following conclusions can be drawn from the research on the HFGVERPTW.

- In order to minimize the total traverse distance and total energy consumption, an optimization model is established that comprehensively considers the AGV structure and motion state, which can more accurately and objectively reflect the actual phenomenon of the workshop intralogistics.
- The GA-LNS considering the farthest insertion heuristic is designed. The improved algorithm by increasing the neighborhood search ability can improve the quality of the initial solution.
- Regarding evolutionary operation, GA-LNS adopts the strategy of the new roulette wheel selection and adaptive mutation operation to ensure the effective convergence and robustness. The effectiveness of algorithm is verified by testing Solomon benchmark instances of different customers scales and comparing with other algorithms.
- The mathematical model established can verify the multitype heterogeneous AGV routing problem with changing motion state and various capacities in reality. This is novel and an expansion of the research on classical VRPTW.
- The results of this study demonstrate that for the current manufacturing context, a heterogeneous AGV fleet transportation method is superior to a homogeneous fleet. Undoubtedly, as the main material transportation equipment inside the FMS, AGV has great energy saving potential.

Consequently, this paper provides a potential basis for future research (1) on integrating the collaborative scheduling of AGV and production machinery into the proposed model; (2) on HFGVRPTW with multiple depots of simultaneous pickup and delivery problems; (3) on how to improve and optimize the performance of GA-LNS; (4) on taking customer satisfaction with the service as an optimization goal; and (5) on considering the coordinated scheduling of multiple distribution centers in FMS.

**Author Contributions:** Conceptualization, X.Z. and J.G.; methodology, X.Z. and J.G.; investigation, X.Z. and J.G.; writing-original draft preparation, X.Z. and J.G.; writing-review and editing, J.G.; supervision, F.G. and Q.H.; and funding acquisition, X.Z. and X.T. All authors have read and agreed to the published version of the manuscript.

**Funding:** This work was supported by the research grant from the Education Department Foundation of Liaoning Province (JDL2019016) and Natural Science Foundation of Liaoning Province (2021-KF-15-02).

**Institutional Review Board Statement:** Excluding this statement.

**Informed Consent Statement:** Excluding this statement.

**Data Availability Statement:** Excluding this statement.

**Conflicts of Interest:** The authors declare no conflict of interest.

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
