# Peer review of "Heterogeneous Multitype Fleet Green Vehicle Path Planning of Automated Guided Vehicle with Time Windows in Flexible Manufacturing System"

_machines, doi:10.3390/machines10030197_

Round 1
Reviewer 1 Report
This paper seemed to be improved greatly by the authors in previous reviewers. In this way, this paper is well-written and well organized. Besides, it presents a good literature review. However, I recommend highlighting the contributions in the introduction (maybe by bullets) and inserting more recent references (i.e., 2020 and 2021). The results must present more qualitative results.
Author Response
Point 1: I recommend highlighting the contributions in the introduction (maybe by bullets) and inserting more recent references (i.e., 2020 and 2021).
Response 1: We are very glad to receive your comments. Undoubtedly, using the bullets to organize manuscript contributions in the introduction part can improve the quality and readability of this article. This improvement will make the the article structure clearer and more intuitive. We have fully considered your suggestion in the process of writing the manuscript, and listed the contribution of this study in line 75 - 85 of this article’s introdution.
Citing more recent references can significantly improve the prospective and forceful of manuscripts. Therefore, in the latest submitted manuscript, we adopted your suggestion and added related references which them are relevant to this research topic within the past two years.
Point 2: The results must present more qualitative results.
Response 2: As suggested by the reviewer, in order to illustrate the veracity and statistical significance of the data presented here, we have included more qualitative analysis results of the experimental data in the latest submitted manuscript.
Reviewer 2 Report
I am satisfied with revision.
Author Response
Point 1: I am satisfied with revision.
Response 1: We appreciate for editor and reviewer’s warm work earnestly, your comments are quite helpful in improving the quality and readability of this manuscript, and help us for the further improvement.
Reviewer 3 Report
Heterogeneous Multi-type Fleet Green Vehicle Path Planning 2
of Automated Guided Vehicle with Time Windows in Flexible 3
Manufacturing System
The problem of vehicle routing is actual and results are very important for mobile robots, smart cars and other engineering areas.
Based on a hybrid genetic algorithm, in present approach, multiple test ware performed in simulation (on Matlab).
Some (possible) short observations:
- maybe, find another style for presentation of Table 1;
- for presented/proposed (two type) vehicles, the speed is constant of variable?
Authors must underline, in conclusions section, the advantages of this solution with respect to other "classic" approaches.
Author Response
Point 1: Maybe, find another style for presentation of Table 1;
Response 1: Based on your suggestion, in the latest submitted manuscript, we have made modest modifications to Table 1.
Point 2: For presented/proposed (two type) vehicles, the speed is constant of variable?
Response 2: In this study, the speed of two type vehicles when going straight and turning is constant value.
Point 3: Authors must underline, in conclusions section, the advantages of this solution with respect to other "classic" approaches.
Response 3: Thanks very much for your comments, which are very helpful to improve the quality of this article. In the latest submission of the revised manuscript, we use bullet points in the article conclusion section to summarize and compare the approach of this study with classical method. At the same time, it is further illustrates the effectiveness of this study.
Reviewer 4 Report
Dear authors,
The article looks corrected and rebuilt according to previous reviewers' recommendations.
My only suggestion is for the reference list. You provide an extensive list of 45 references, but I haven't found a single publication from the journal you want to publish in. You could potentially consider adding any publications from Machines. However, I don't think it is compulsory; it's instead a suggestion for future work.
Best regards,
Author Response
Point 1: Dear authors, The article looks corrected and rebuilt according to previous reviewers' recommendations. My only suggestion is for the reference list. You provide an extensive list of 45 references, but I haven't found a single publication from the journal you want to publish in. You could potentially consider adding any publications from Machines. However, I don't think it is compulsory; it's instead a suggestion for future work.
Response 1: We reviewed recent articles published in Machines journal which are related to the topic of this study, and we have revised and replaced the previous references list. In the latest submitted revised manuscript, the revised references list contains 5 recent articles pulished by Machines. Thanks again for your suggestions!
This manuscript is a resubmission of an earlier submission. The following is a list of the peer review reports and author responses from that submission.
Round 1
Reviewer 1 Report
Rewrite the abstract to be more direct and concise. Besides, insert more info about the results in the abstract;
The introduction must present the motivation to the problem that is being solved. Insert the difficulties faced by heterogeneous multi-type fleet AGVs;
Summarize through bullets the main contributions of the manuscript;
Insert more recent references in the section Literature Review and add a table comparing some characteristics of your proposition and similar ones from the literature;
Table 1 should be presented at the beginning of the manuscript or at the beginning of section 3 (before the problem formulation);
In order to help the understanding of the readers, please, insert a figure with a flowchart explaining the proposition in the problem formulation section;
More qualitative results should be added. Besides, the authors should insert statistical analysis.
The authors should rewrite or better organize the results section. It is confusing in the way that is presented.
Figures must be of good quality. The text inside them should be readable. Fix the figures 1,4, 6, 7 e 8;
The conclusions must be more concise and direct. Insert more ideas of future works,
minor revisions:
- Check the English grammar;
- tables and figures should be in the same size as the text of the manuscript;
- Insertion of space after some worlds that have acronyms explained (e.g., in the abstract), such as in "vehicle routing problem(VRP)," heterogeneous fleet VRP(HFVRP)," etc. Other words in the manuscript have this same problem, such as in line 203 "Node0"
Reviewer 2 Report
heterogeneous multi-type fleet green vehicle routing problem with time windows(HFGVRPTW) is studied in this paper.
Using experiments, computational load and energy consumption of your method needs to be analyzed and compared with the-state-of-the-art.
Energy consumption analysis needs to be performed using experiments. I think the load weight can increase the energy consumption of the robot carrying the load.
Assumption (4) in Section 3.1 is too strong.
"Once the material handling process assigned, it will be ignored traffic problems, collision, conflict, or deadlock" This assumption is too strong.
As multiple robots move, traffic problem, collision, conflict must be considered.